# miRNA Mediated Noise Making of 3′UTR Mutations in Cancer

**DOI:** 10.3390/genes9110545

**Published:** 2018-11-12

**Authors:** Wei Wu, Lingxiang Wu, Mengyan Zhu, Ziyu Wang, Min Wu, Pengping Li, Yumin Nie, Xue Lin, Jie Hu, Eskil Eskilsson, Qh Wang, Jiaofang Shao, Sali Lyu

**Affiliations:** 1Department of Bioinformatics, Nanjing Medical University, Nanjing 211166, China; wuwei.bioinformatics@gmail.com (W.W.); ecsivo@gmail.com (L.W.); zhumengyan1994@njmu.edu.cn (M.Z.); wzy_njmu@outlook.com (Z.W.); biowumin@gmail.com (M.W.); lipengping@njmu.edu.cn (P.L.); yumin_nie@njmu.edu.cn (Y.N.); xue.lin@njmu.edu.cn (X.L.); hujie@njmu.edu.cn (J.H.); 2Independent, San Francisco, CA 94952, USA; eskileskilsson@gmail.com; 3Key Laboratory of Human Functional Genomics of Jiangsu Province, Nanjing 211166, China; 4Collaborative Innovation Center for Cardiovascular Disease Translational Medicine, Nanjing 211166, China; 5Jiangsu Key Lab of Cancer Biomarkers, Prevention and Treatment, Collaborative Innovation Center for Cancer Personalized Medicine, Nanjing 211166, China; 6State Key Laboratory of Reproductive Medicine, Nanjing Medical University, Nanjing 211166, China

**Keywords:** miRNA polymorphism, drug response alteration, drug target, functional somatic mutation, gene expression

## Abstract

Somatic mutations in 3′-untranslated regions (3′UTR) do not alter amino acids and are considered to be silent in cancers. We found that such mutations can promote tumor progression by altering microRNA (miRNA) targeting efficiency and consequently affecting miRNA–mRNA interactions. We identified 67,159 somatic mutations located in the 3′UTRs of messenger RNAs (mRNAs) which can alter miRNA–mRNA interactions (functional somatic mutations, funcMutations), and 69.3% of these funcMutations (the degree of energy change > 12 kcal/mol) were identified to significantly promote loss of miRNA-mRNA binding. By integrating mRNA expression profiles of 21 cancer types, we found that the expression of target genes was positively correlated with the loss of absolute affinity level and negatively correlated with the gain of absolute affinity level. Functional enrichment analysis revealed that genes carrying funcMutations were significantly enriched in the MAPK and WNT signaling pathways, and analysis of regulatory modules identified eighteen miRNA modules involved with similar cellular functions. Our findings elucidate a complex relationship between miRNA, mRNA, and mutations, and suggest that 3′UTR mutations may play an important role in tumor development.

## 1. Introduction

MicroRNAs (miRNAs) are a class of small noncoding RNAs consisting of 19–25 nucleotides that play important roles in regulating mRNA expression at the post-transcriptional level. By binding to 3′-untranslated regions (3′UTR) of cytosolic messenger RNAs (mRNAs), miRNAs either reduce the translation or increase the degradation of transcripts [1]. Cumulative studies have demonstrated the importance of miRNAs in targeting pivotal genes to promote tumor progression [2,3,4].

Somatic mutations play crucial rules in tumor initiation and progression [5,6,7]. A small number of mutations are identified to be “driver mutations” that yield a selective clonal growth advantage to cancer cells. Meanwhile, the large majority of mutations are “passenger mutations” which do not explicitly yield a growth advantage but may nonetheless provide useful information about the evolutionary forces within the cancer genome [8]. Studies have demonstrated that the accumulation of somatic mutations influence the regulation of several important signaling pathways including the P53, TGF-β, and WNT signaling pathways [9,10]. Importantly, different mutational processes may generate different mutation types (or signatures) [11]. For instance, the UV-induced mutation was strongly associated with C>T and CC>TT alterations [12], and it was recently suggested that temozolomide (TMZ) treatment may also increase the C>T alterations in glioblastoma multiforme (GBM) patients [13].

Non-synonymous gene mutations which alter the amino acids of protein products may result in significant functional changes and accelerate the progression of tumors. For example, somatic mutation at R132 of *IDH1*, a gene encoding NADP+-dependent isocitrate dehydrogenase 1, was found to be associated with early gliomagenesis [14]. By changing the function of the enzyme to produce 2-hydroxyglutarate, the *IDH1* mutation remodeled the environment to fuel tumor cell development [15]. Synonymous mutations, meanwhile, do not alter amino acids and are largely considered to be “silent mutations”. Yet, Supek et al. present a compelling analysis that suggests synonymous mutations in cancer can be oncogenic by altering transcript splicing and thereby affecting protein function [16]. Others have also implicated mutations in cancer [17].

Akdeli et al. found a 3′UTR polymorphism that can influence *PLK1* mRNA stability and may be a factor for response to *PLK1* inhibitors [18], suggesting that the mutations residing in the 3′UTR may alter miRNA binding efficiency and consequently trigger loss/gain of gene function. Cui et al. found 11 mutations in 3′UTR that alter miRNA target sites in cancer-related genes [19]. A number of methods and tools have been developed for compiling the compendiums of functional genomic variations in miRNA target sites, including PolymiRTS [20], Patrocles [21], MicroSNiPer [22], and dbSMR [23]. SomamiR is a database for collecting somatic mutations in miRNAs and their target sites that potentially alter the interactions between RNAs [24]. Systematic characterization of the functional implications of mutations in cancers remains to be done.

In the present study we analyzed 744,270 3′UTR mutations among which we identified 67,159 mutations located in the 3′UTR of mRNAs that can alter miRNA–mRNA interactions, and 69.3% of these functional somatic mutations (funcMutations, the degree of energy change > 12 kcal/mol) were found to result in the loss of miRNA-mRNA binding. G>T mutations were observed with the highest proportion (about 22.9%) among funcMutations. The C and G wild-type mutation was prone to result in loss of miRNA binding while the A and T wild-type mutation may promote the ability for genes to bind with miRNAs. By integrating mRNA expression profiles of 21 cancer types, we found that the expression of target genes showed a positive correlation with the loss of absolute affinity level and a negative correlation with the gain of absolute affinity level. Functional enrichment analysis revealed that the funcMutations may regulate gene expression and affect cancer-related signaling pathways (e.g., WNT signaling pathway). Analysis of regulatory modules identified eighteen miRNA modules co-regulating similar cellular processes. Our findings suggest that 3′UTR mutations may perturb miRNA–mRNA interactions and consequently have important implications for tumor development.

## 2. Materials and Methods

### 2.1. Functional Somatic Mutation Identification

Sequences of mature miRNA were acquired from miRBase version 21 [25]. Somatic mutations located in the 3′UTRs were obtained from The Cancer Genome Atlas (TCGA, https://portal.gdc.cancer.gov, annotated data) [26]. Variants calling results (VCF files) of whole-exome target sequencing (WES) datasets of 10,429 tumors for 33 cancer types were obtained from TCGA which detected 1,648,474 potential single nucleotide variants in 3′UTRs. The 3′UTR position annotation analysis for mutation data were conducted by reference file derived from Ensembl (https://www.ensembl.org/index.html, Version GRCh38.p10) [27]. The sequences of mRNA 3′UTRs were obtained from the UCSC Genome browser (UCSC Genome Browser, http://genome.ucsc.edu/, version: GRCh38, Dec. 2013) [28]. The mRNAs with or without 3′UTR mutations were considered as the mutant or the wild-type mRNA, respectively. The sequences of mRNAs (either mutant or wild-type) were extended 30 bp on both upstream and downstream of variant position. Finally, these paired sequences were analyzed by Probability of Interaction by Target Accessibility (PITA) [29] (with default parameters) to assess the change of free energy and predict potential miRNA target sites. We defined that miRNA and mRNA had a binding event if the absolute value of the PITA score > 10 kcal/mol (ΔΔG < −10). The somatic mutations changing the free energy between miRNA and mRNA sequences were defined as functional somatic mutations (funcMutations). According to the binding of miRNA to the mRNA 3′UTR, we assigned the potential funcMutations to one of the four classes: “complete gain”, the mRNA acquires a new miRNA binding site through the wild-type somatic mutation into variant-type somatic mutation; “complete loss”, mRNA loses a predicted miRNA binding site through the wild-type somatic mutation into variant-type somatic mutation; “partial gain”, mRNA acquires more stable miRNA binding site than that without the somatic mutation; “partial loss”, mRNA target site turns into unstable miRNA binding site with the somatic mutation. The degree of binding is quantified by the change of the PITA score, which was defined as follows:$$\mathrm{absolute}\;\mathrm{affinity}=\left|T_{var}-T_{wt}\right|$$
where $T_{wt}$ represents the score of miRNA binding to the wild-type 3′UTR sequences and $T_{var}$ represents the score of miRNA binding to the variant-type 3′UTR sequences. Absolute affinity represents a strengthened miRNA regulation ability from wild-type allele to the variant-type. To analyze the position feature of somatic mutations in 3′UTR of genes, we calculated the distance between somatic mutations and the start position of 3′UTR of genes and the length of the localized 3′UTR of genes. Finally, we calculated the relative position of somatic mutations in 3′UTR.

### 2.2. Gene Expression Analysis and Tumor Hallmarks Enrichment Analysis

mRNA expression data of multiple cancer types were obtained from TCGA (RNA-SeqV2). The expression profiles were normalized level 3 data across twenty-one kinds of cancer types (total 6078 samples), including bladder urothelial carcinoma (BLCA), breast invasive carcinoma (BRCA), cervical squamous cell carcinoma and endocervical adenocarcinoma (CESC), cholangiocarcinoma (CHOL), colon adenocarcinoma (COAD), esophageal carcinoma (ESCA), head and neck squamous cell carcinoma (HNSC), kidney chromophobe (KICH), kidney renal clear cell carcinoma (KIRC), kidney renal papillary cell carcinoma (KIRP), liver hepatocellular carcinoma (LIHC), lung adenocarcinoma (LUAD), lung squamous cell carcinoma (LUSC), pancreatic adenocarcinoma (PAAD), pheochromocytoma and paraganglioma (PCPG), prostate adenocarcinoma (PRAD), rectum adenocarcinoma (READ), stomach adenocarcinoma (STAD), thyroid carcinoma (THCA), thymoma (THYM), and uterine corpus endometrial carcinoma (UCEC). In order to reduce the influence of the batch effect of expressions across different cancer types, the gene expression value was adjusted by the average expression within the same cancer type. The ratio for each gene was denoted as the ratio of the gene expression in mutated sample to the mean of gene expression level in all samples.

To interpret the influence of functional somatic mutations on tumors, we applied the affected genes (designated as tarGenes) to enrichment analysis, including subcellular localization (from The Human Protein Atlas, https://www.proteinatlas.org/) [30,31] and KEGG pathway enrichment analysis. In addition, tumor hallmarks (50 gene sets) were available from GSEA dataset (Gene Set Enrichment Analysis, http://software.broadinstitute.org/gsea/index.jsp) [32,33] and the relationship between tarGenes and cancer hallmarks were explored. The network was visualized by Cytoscape (https://www.cytoscape.org) [34].

## 3. Results

### 3.1. Identification of Functional Somatic Mutations in 3′UTRs

We obtained a total of 2048 sequences of miRNA from miRBase and a total of 25,539 effective 3′UTR sequences of transcripts of human protein-coding genes from the UCSC Genome browser. We integrated 744,270 somatic mutations located in 3′UTR from TCGA. Through mapping somatic mutations into 3′UTR of transcripts, 451,002 out of 744,270 somatic mutations were mapped into the 3′UTR sequences of transcripts obtained from the UCSC Genome browser. For each somatic mutation in human mRNA 3′UTR regions, we assessed whether its two alleles would cause different miRNA-mRNA target binding. Then, we obtained 451,002 pairs of the length of 61 bp of wild-type and variant-type mRNA sequences. After calculating the PITA score, 67,159 funcMutations were detected and these somatic mutations causing the change of 3,314,904 pairs of miRNA-mRNA binding, as shown in Appendix A. In addition, 9107 funcMutations were selected for subsequent analysis, which most significantly impacted 33,385 pairs of miRNA–mRNA interactions and the absolute affinity (degree of energy change) > 12 kcal/mol, as shown in Appendix A. On the other hand, we integrated and compared the results with experimentally validated miRNA–mRNA interactions found in miRTarBase [35]. We found that 54,244 of 3,314,904 pairs of miRNA–mRNA interactions were experimentally validated, as shown in Appendix A.

### 3.2. Position Feature and Base Substitution Characteristics of 3′UTR Mutations

In the present study, all mutations located in the 3′UTRs were collected from TCGA. To explore the positional features of these somatic mutations, we conducted a statistical analysis of the positional distribution of somatic mutations in 3′UTR. Intriguingly, the position of somatic mutations concentrated on 5′ ends of 3′UTR, which is closer to the gene body, as shown in Figure 1A. This finding may be caused by data bias as nearly all samples are WES data. The Illumina TruSeq Exome Enrichment Kit (Illumina, San Diego, CA, USA) is used by the TCGA project and covers 62 Mb DNA sites. Somatic mutations located in 3′UTR can be captured by WES since the enriched sequences include ~28 Mb of UTR. In addition, somatic mutations in TCGA cases were identified by Mutect2, which is a sensitive method to detect somatic mutations. Mutect2 will provide at least 80% power to detect mutations with an allelic fraction of 0.3 if the coverage of depth is more than 14-fold in tumors and 8-fold in normal samples [36,37]. In order to confirm whether the capture kit for WES can provide efficient power to detect mutations in 3′UTR, we analyzed the coverage of depth of 3′UTR in both tumor and normal samples. We found that 85.3% of 3′UTRs have more than 14-fold coverage in tumors, as shown in Appendix A, and 88.5% of 3′UTRs have more than 8-fold coverage in normal samples, as shown in Appendix A. The analysis demonstrates that the coverage of depth captured by the targeted sequencing kit utilized in the TCGA project provides an effective capability to detect mutations in 3′UTR.

To assess the functional implications of different base substitutions located in the 3′UTR of genes, we analyzed the different base substitutions and their effect on binding status. We found that G>T mutations (~22.9%) occurred most frequently. Intriguingly, C and G wild-type mutations tended to result in loss of miRNA binding. Meanwhile, A and T wild-type mutations were more likely to result in gain of miRNA binding, as shown in Figure 1B. Among the 30 mutational signatures collected in COSMIC (Catalogue of Somatic Mutations in Cancer, https://cancer.sanger.ac.uk/cosmic/) [38], signatures 11 and 23 exhibit strong transcriptional strand-bias for C>T mutation. For example, signature 11 has been found in melanomas and glioblastomas and exhibits a mutational pattern resembling that of alkylating agents. Patient histories have revealed an association between treatment with the alkylating agent TMZ and Signature 11 mutations [13]. Accordingly, we speculate that some mutations can impact cancer development by altering miRNA targeting efficiency and thereby affecting miRNA-mRNA binding status.

### 3.3. The Impact of 3′-Untranslated Regions Mutations on Gene Expression

The function of a gene depends on its expression and may be regulated by miRNAs. To further explore the implications of 3′UTR mutations for gene expression, we integrated mRNA expression across 21 cancer types (normalized by average expression) and analyzed their funcMutations. We focused on the genes having gain or loss of miRNA binding for which the frequency detected in cancer types was greater than two (common mutation) and the average absolute affinity was greater than 1 kcal/mol (explicit loss or gain status). As expected, we found that the expression of tarGenes exhibited a positive correlation with the loss of absolute affinity level, as shown in Figure 2A. In contrast, in gaining status, expression of tarGenes significantly declined with the increase of absolute affinity level, as shown in Figure 2B. These findings demonstrate the importance of funcMutations on the regulation of gene expression.

### 3.4. 3′UTR Mutations Promote Tumor Progression

To evaluate the influence of 3′UTR mutations in tumors, we performed enrichment analysis of the tarGenes by using subcellular localization and KEGG pathway enrichment analysis. Most of the organelles were enriched with a slight proportion (~0.3) and only five were found with a proportion ranging from 0.33 to 0.50. Moreover, Fisher’s exact test analysis revealed that tarGenes were significantly correlated with expression in the cytosol, vesicles, nucleoplasm, nucleus, mitochondria, and nucleoli (Benjamini–Hochberg adjustment, q-value <$10^{-3}$), as shown in Figure 3A. The enrichment analysis based on the KEGG pathway indicated that tarGenes have a number of cancer-related functions including Pathways in cancer, Basal cell carcinoma, MAPK signaling pathway, Calcium signaling pathway, WNT signaling pathway, Ras signaling pathway, and Rap1 signaling pathway, as shown in Figure 3B.

We also explored the relationship between tarGenes and cancer hallmarks. As expected, the number of tarGene mutations decreased with increasing number of hallmarks, as shown in Figure 4A. By leveraging permutation analysis, we selected some of the tarGenes which exhibited large numbers of mutations relative to genes involved in an equal number of hallmarks, as shown in Appendix A, as we speculated they may be important for tumor progression. For example, *PRX* (permutation q-value < 0.001, mutations = 18) is a protein coding gene that contains PSD95 (post synaptic density protein) and DlgA (Drosophila disc large tumor suppressor) domains. The 3′UTR mutations of *PRX* generally resulted in a loss of binding affinity to miRNAs, as shown in Figure 4B. The miRNAs may otherwise up-regulate *PRX* expression and thereby inhibit tumor progression. *HIPK2* (permutation q-value = 0.001, mutations = 10), a tumor suppressor gene [39], was found to be involved in two tumor hallmarks. The 3′UTR mutations in *HIPK2* generally resulted in a gain of binding affinity to miRNAs, as shown in Figure 4C. The miRNAs may down-regulate the expression of *HIPK2* and thereby promote tumor progression. The 3′UTR mutations in *PPARD* (permutation *p*-value = 0.001, mutations = 10) were associated with loss of miRNA binding, as shown in Figure 4D. The binding of miRNAs may otherwise inhibit *PPARD* expression as studies have demonstrated that *PPARD* expression promotes tumor progression and metastasis [40]. These findings together indicate that mutations in the 3′UTR of tarGenes may have a complex impact on tumor progression by changing binding affinity of a spectrum of miRNAs.

### 3.5. microRNA Modules Co-Regulate the Tumor Hallmarks

Based on the analysis above, we next explored the involvement of miRNAs in co-regulating tumor hallmarks. In general, the 3′UTR mutations in cancers contributed to a loss of binding affinity (69.3% of the funcMutations were identified triggering the loss of miRNA-mRNA binding), as shown in Figure 5A. To explore the function of changes in binding ability (designated as a loss and gain status), we applied the tumor hallmark enrichment analysis of the corresponding tarGenes to each of the miRNAs. In the loss status, a number of potential tumor suppressor miRNAs were free from mutated binding sites, including hsa-miR-4763-3p, and hsa-miR-328-5p [41]. Meanwhile, tumor progression hallmarks that support the invasion and metastasis of tumors, including inflammatory response, DNA repair, and IL6/JAK/STAT3 signaling, were activated, as shown in Figure 5B. A number of miRNAs (e.g., hsa-miR-6803-5p, etc.) that re-gained the tarGene binding site may inhibit pathways otherwise involved in tumor suppression (e.g., the P53 pathway) or activate pathways involved with oncogenesis (e.g., the IL6/JAK/STAT3 pathway), as shown in Figure 5C. We investigated the latent correlation between the miRNAs that gained binding ability to new tarGenes (e.g., the number of tarGenes > 100). By conducting Fisher’s exact test (q-value < 0.01, odds ratio > 10) and MCODE cluster analysis (default parameter), we found eighteen modules consisting of 98 miRNAs, as shown in Figure 5D and Appendix A. For example, module3 was derived from the miRNAs having similar functions. Previous studies have demonstrated that the hsa-miR-331-3p [42], hsa-miR-4486 [43], hsa-miR-378g [44], hsa-miR-6860 [45], hsa-miR-615-3p [46], and hsa-miR-5189-5p [47] were involved in the proliferation, invasion, and metastasis of cancers. Herein we provide a new method for analysis of miRNA modules co-regulating tumor pathways, and our findings contribute to a better understanding of the implications of 3′UTR mutations in tumor biology.

## 4. Discussion

Mutations of 3′UTR may affect tumor progression. In this article, we analyzed the mutation position feature, base substitution characteristics, tumor progression, and gene regulation to explore a broad landscape of miRNA–mRNA interactions. Theoretically, the greater change of free energy between miRNA and mRNA sequences indicates an increasing or decreasing impact of somatic mutations on miRNA-mRNA binding which may, in turn, result in a negative or positive regulation of mRNA expression. Most miRNAs primarily target the 3′UTR regions of mRNAs, and accordingly, the present study focused on that aspect in an effort to promote confidence of the analysis results. Future studies will integrate more functional somatic mutations in different regions of both miRNA and their target sites to further explore the influence of mutations on miRNA-mediated gene regulation.

To explore how mutations may affect tumor progression, we assessed the position feature and base substitution characteristics of 3′UTR mutations. We found that some 3′UTR mutations alter miRNA targeting efficiency and thereby can impact tumor development. By analyzing the position feature, however, we found that sequencing technology may introduce data bias. In future studies, we need unbiased whole genome sequencing data to confirm our findings. To further evaluate the influence of mutations in tumors, we next integrated subcellular localization and KEGG pathway enrichment analysis to functionally explore the genes with 3′UTR somatic mutations. The analysis revealed significant enrichment of a number of cancer-related processes including MAPK and WNT signaling pathways. In addition, we identified that regulation of some of the tarGenes (e.g., *PRX*, *HIPK2*, and *PPARD*) may be important for tumor progression. Overall, the differential analysis demonstrated that the 3′UTR mutations in cancers generally caused an increasing loss of miRNA binding ability. The miRNAs with either loss or gain status appeared to regulate a series of tarGenes to activate or deactivate the tumor-related pathways in a parallel fashion which, together, may result in coordinated and stable effects in tumor development. Specifically, we identified eighteen miRNA modules co-regulating similar functions. We also found that the expression of tarGenes positively correlated with the loss of absolute affinity level and negatively correlated with the gain of absolute affinity level in 21 cancer types. Our findings together demonstrate that 3′UTR mutations may play an important role in tumor development and the phenomenon should be further investigated.

## Figures and Tables

**Figure 1 genes-09-00545-f001:**
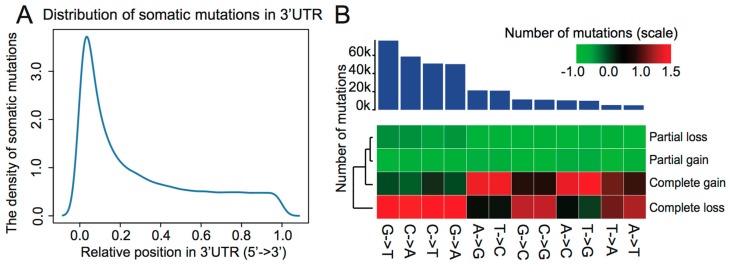
(**A**) The distribution of somatic mutations in 3′-untranslated regions (3′UTR). Horizontal axis represents the relative position of somatic mutations in 3′UTR from 5′ to 3′. Vertical axis represents the density of somatic mutations located in 3′UTR. (**B**) Contributions of base substitutions in 3′UTR mutations.

**Figure 2 genes-09-00545-f002:**
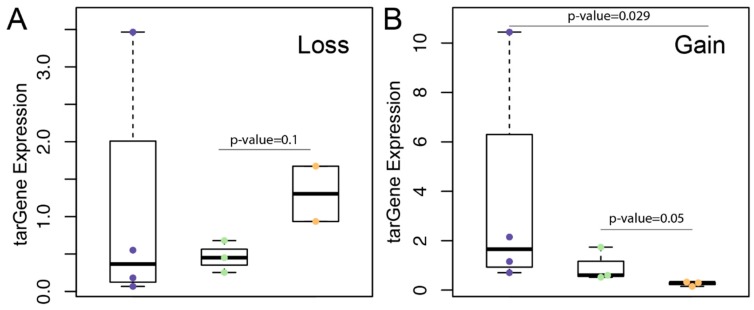
(**A**,**B**) Correlation between the expression of tarGene and miRNA (microRNA) binding absolute affinity (**A**: loss status, **B**: gain status). Vertical coordinate represents the ratio of the gene expression in mutated sample to the mean of gene expression level in all samples. Three colors represent different degrees of the absolute affinity level. Statistical testing was performed using the Wilcoxon rank-sum test.

**Figure 3 genes-09-00545-f003:**
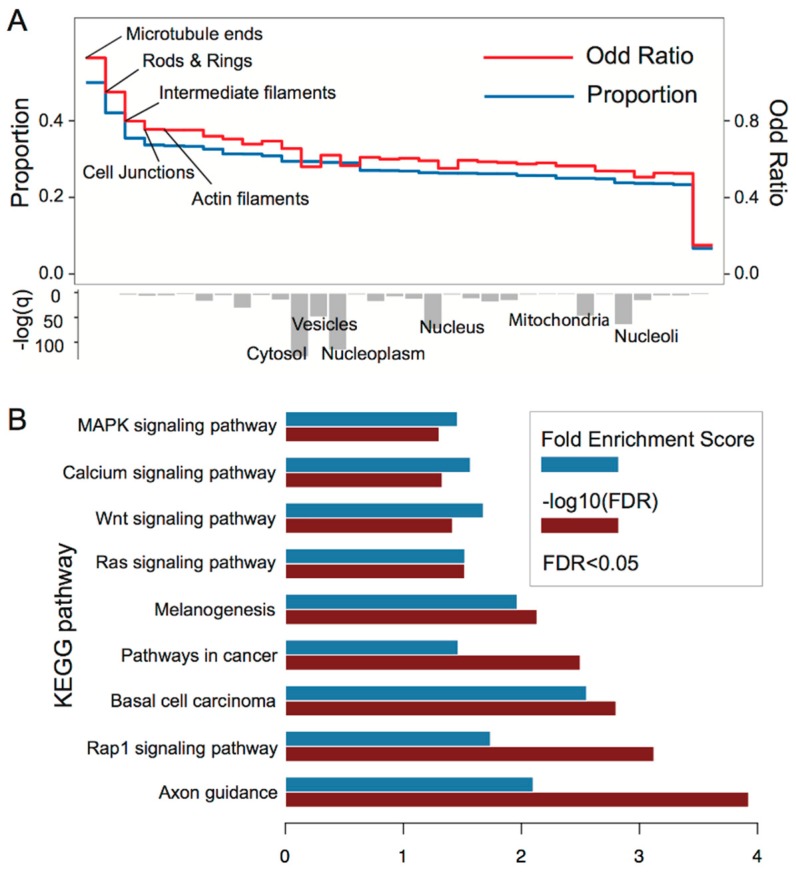
(**A**) Subcellular localization. Blue line denotes the hit proportion of tarGenes across subcellular gene sets. Red line signifies the odd ratio of tarGenes across subcellular gene sets. The bottom bar plot represents the log-transformed q-value (Benjamini–Hochberg adjustment) for enrichment analysis of each subcellular gene set. (**B**) KEGG pathway enrichment analysis, the false discovery rate <0.05.

**Figure 4 genes-09-00545-f004:**
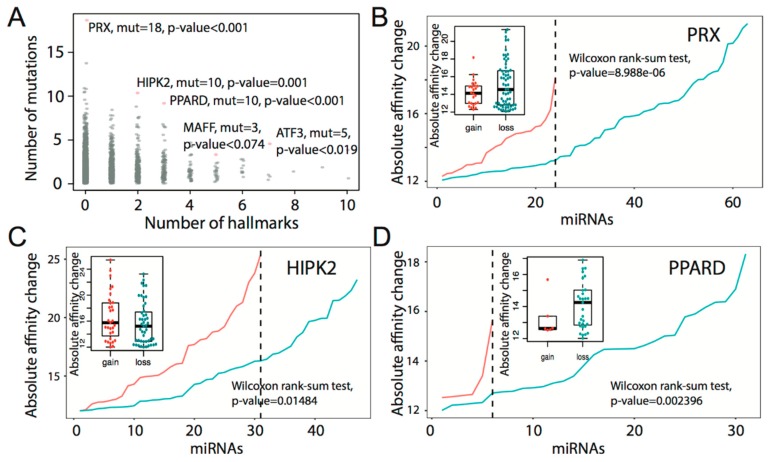
(**A**) Number of hallmarks versus number of mutations across tarGenes. Each tarGene was subjected to permutation analysis for significance evaluation. (**B**–**D**) Differential analysis of affinity changes between loss and gain of 3′UTR mutation in PRX, HIPK2, and PPARD. Blue and red lines represent the absolute affinity change of loss and gain, respectively.

**Figure 5 genes-09-00545-f005:**
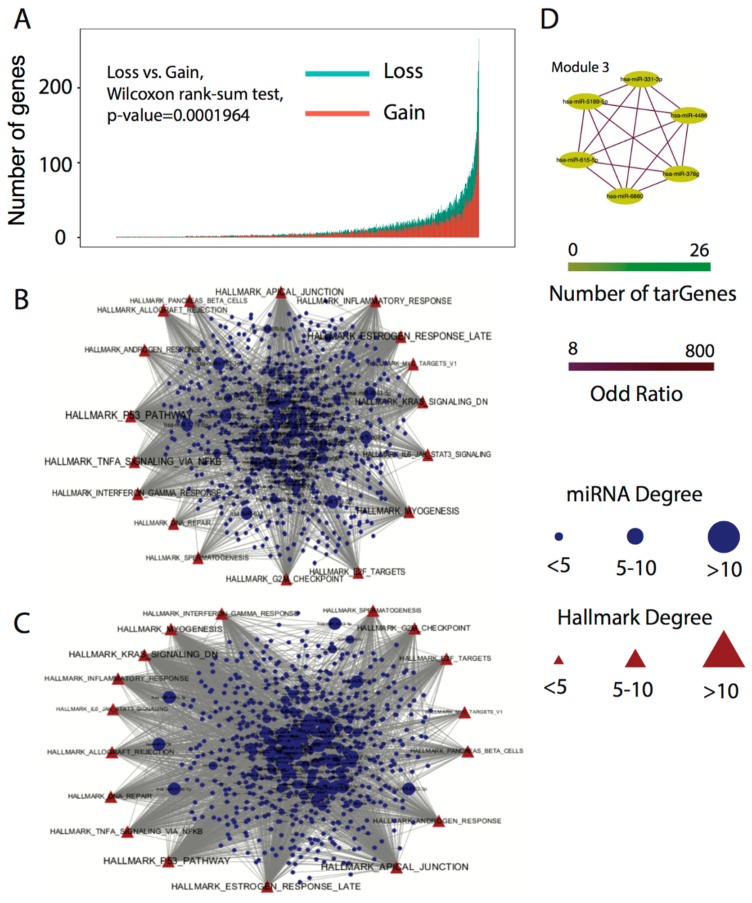
(**A**) Influence of miRNAs on genes. Red bar represents the number of genes bound by miRNA which re-gained binding sites. Blue bar represents the number of genes that were free from miRNA due to the loss of binding sites; (**B**) Connections between tumor hallmarks and miRNAs in the loss status; (**C**) Connections between tumor hallmarks and miRNAs in the gain status; (**D**) The module of miRNAs in gain status.

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
