# Peer review of "miRNA Mediated Noise Making of 3′UTR Mutations in Cancer"

_genes, 2018, doi:10.3390/genes9110545_

Round 1
Reviewer 1 Report
Wu et al. are presenting a comprehensive analysis of somatic mutations in tumors and how they affect miRNA binding sites. I only have the following minor points:
a) Authors write they "intergated 744,270 somatic mutations in 3'UTR from TCGA" but in the next senetence they claim that by mapping mutations into 3'UTR transcripts they found 451,002 mutations in 3'UTR. Why this difference in numbers occur is not clear and confusing.
b) In the comparison between the computationally acquired mRNA-miRNA pairs and the functionally validated list, what is the overlap of the 33,585 and 54,244 pairs? As the text reads now it seems like 54,244 of 33,585 pairs were validated, which makes no sense at it equates to more than 100%.
c) I suggest authors replace "silent mutations" with mutations. Silent mutations are by definition mutations that have no effect on the phenotype, hence wording such as "silent mutations may have an important role in tumor development" is an oxymoron.
d) Figure 1 is very interesting, but I see no point in distinguishing transcripts on positive and negative strand unless the authors believe there is a reason why they could be different. But they are silent on any such hypothesis.
Author Response
Response to Reviewer 1 Comments:
Wu et al. are presenting a comprehensive analysis of somatic mutations in tumors and how they affect miRNA binding sites. I only have the following minor points:
Point 1: Authors write they "intergated 744,270 somatic mutations in 3'UTR from TCGA" but in the next senetence they claim that by mapping mutations into 3'UTR transcripts they found 451,002 mutations in 3'UTR. Why this difference in numbers occur is not clear and confusing.
Response 1: We apologize for the unclear description in the manuscript. A total of 744,270 somatic mutations located in the 3’UTRs were obtained from TCGA (https://portal.gdc.cancer.gov). However, only 451,002 out of the 744,270 mutations were mapped within the 3’UTRs of annotated genes in the GRCh38 genome derived from UCSC Genome browser (http://genome.ucsc.edu/). These mutations (with sequence information) were finally used for calculating PITA scores. We revised the manuscript as follows:
Line 143-146: “We integrated 744,270 somatic mutations located in 3’UTR from TCGA. Through mapping somatic mutations into 3’UTR of transcripts, 451,002 out of 744,270 somatic mutations were mapped into the 3’UTR sequences of transcripts obtained from UCSC Genome browser.”
Point 2: In the comparison between the computationally acquired mRNA-miRNA pairs and the functionally validated list, what is the overlap of the 33,585 and 54,244 pairs? As the text reads now it seems like 54,244 of 33,585 pairs were validated, which makes no sense at it equates to more than 100%.
Response 2: Many thanks to the reviewer for pointing out the mistake. We have revised the manuscript according to the comment as follows:
Line 149-157: “After calculating PITA score, 67,159 funcMutations were detected and these somatic mutations causing the change of 3,314,904 pairs of miRNA-mRNA binding (Supplement Table S1). In addition, 9107 funcMutations were selected for subsequent analysis, which most significantly impact 33,385 pairs of miRNA-mRNA interaction and the absolute affinity (degree of energy change) > 12 kcal/mol (Supplement Figure S1). On the other hand, we integrated and compared the results with experimentally validated miRNA-mRNA interaction found in miRTarBase [28]. We found that 54,244 of 3,314,904 pairs of miRNA-mRNA interaction were experimentally validated (Supplement Table S2).”
Point 3: I suggest authors replace "silent mutations" with mutations. Silent mutations are by definition mutations that have no effect on the phenotype, hence wording such as "silent mutations may have an important role in tumor development" is an oxymoron.
Response 3: Many thanks to the reviewer for pointing this out. We have revised the manuscript by replacing all the "silent mutations" with “mutations”.
Point 4: Figure 1 is very interesting, but I see no point in distinguishing transcripts on positive and negative strand unless the authors believe there is a reason why they could be different. But they are silent on any such hypothesis.
Response 4: Many thanks to the reviewer for pointing this out. We have revised the Figure 1A without distinguishing transcripts on positive and negative strand. We revised the manuscript as follows:
Line 188-191:
Figure 1. (A) The distribution of somatic mutations in 3’UTR. Horizontal axis represents relative position of somatic mutations in 3’UTR from 5’ to 3’. Vertical axis represents the density of somatic mutations located in 3’UTR. (B) Contributions of base substitutions in 3’UTR mutations.
Reviewer 2 Report
In manuscript entitled " miRNA Mediated Noise Making of 3’UTR Silent 3 Mutations in Cancer", Wu et al. analyzed TCGA cancer data based on an interesting hypothesis that silent mutations on mRNA 3'UTRs can alter targeting efficiency of miRNAs, which are important for tumor development. With sound analyses on the position feature, base substitution characteristics, tumor progression, gene regulation and functional signaling pathways, the authors revealed that somatic mutations in 3'UTR have an effect on miRNA-mRNA interaction, which in turn regulates mRNA expression and subsequent promotes tumorigenesis in tumor. The methodology and analyses in this manuscript were described clearly; the findings sound very interesting, which may provide new insights for understanding of tumor biology and therapy. Below are several comments/criticisms of this work:
1. MiRNA binding activity is known to be dosage specific dynamics. Why do the authors use all annotated miRNAs and mRNAs (2,048 and 25,539, respectively), instead of expressed ones?
2. To support why miRNA and mRNA is defined to have binding if PITA score > 10kcal/mol. In other words, are there other papers using the same definition or other rationale allowing them to make the definition?
3. It is unclear how they drew the conclusion that silent mutations contributed to a loss of binding affinity from Fig. 5A.
Author Response
Response to Reviewer 2 Comments:
In manuscript entitled " miRNA Mediated Noise Making of 3’UTR Silent 3 Mutations in Cancer", Wu et al. analyzed TCGA cancer data based on an interesting hypothesis that silent mutations on mRNA 3'UTRs can alter targeting efficiency of miRNAs, which are important for tumor development. With sound analyses on the position feature, base substitution characteristics, tumor progression, gene regulation and functional signaling pathways, the authors revealed that somatic mutations in 3'UTR have an effect on miRNA-mRNA interaction, which in turn regulates mRNA expression and subsequent promotes tumorigenesis in tumor. The methodology and analyses in this manuscript were described clearly; the findings sound very interesting, which may provide new insights for understanding of tumor biology and therapy. Below are several comments/criticisms of this work:
Point 1: MiRNA binding activity is known to be dosage specific dynamics. Why do the authors use all annotated miRNAs and mRNAs (2,048 and 25,539, respectively), instead of expressed ones?
Response 1: Many thanks to the reviewer for the comment. It is a challenge for completely dissecting the relationship between dosage specific dynamics and miRNA binding activity due to the complexity of miRNA-mRNA interaction network and the diversity of expression distribution of miRNAs between samples. To cover all potential miRNA binding events as far as possible, we used all annotated miRNAs and mRNAs. Our study hence leveraged a wider exploration of miRNA binding activity caused by the 3’UTR mutations. And again, thanks for the advice, we will systematically explore the miRNA and mRNA expression distribution across different cancer types and unravel the dynamic regulation network consisting of miRNAs and mRNAs in our further study.
Point 2: To support why miRNA and mRNA is defined to have binding if PITA score > 10kcal/mol. In other words, are there other papers using the same definition or other rationale allowing them to make the definition?
Response 2: Many thanks to the reviewer for pointing this out. For the cutoff selection, the official website of PITA explained as follows:
“Since ΔΔG is an energetic score, the lower (more negative) its value, the stronger the binding of the microRNA to the given site is expected to be. As a rough rule of thumb, sites having ΔΔG values below -10 are likely to be functional in endogenous microRNA expression levels. However, given that the actual level of repression depends on microRNA concentration, even sites having ΔΔG values above -10 may be functional at high microRNA expression levels.” (https://genie.weizmann.ac.il/pubs/mir07/mir07_notes.html).
In addition, accumulative studies used the same definition, such as PMID: 25329664 and PMID: 20413099. In our manuscript, we defined that miRNA and mRNA had a binding event if the absolute value of PITA score > 10 kcal/mol. To avoid inaccuracy, we revised the manuscript as follows:
Line 94-97: “Finally, these paired sequences were analyzed by PITA [27] (with default parameters) to assess the change of free energy and predict potential miRNA target sites. We defined that miRNA and mRNA had a binding event if the absolute value of PITA score > 10 kcal/mol (ΔΔG < -10).”
Point 3: It is unclear how they drew the conclusion that silent mutations contributed to a loss of binding affinity from Fig. 5A.
Response 3: Many thanks to the reviewer for the comment. In this manuscirpt, we found that there were 69.3% of the funcMutations identified triggering the loss of miRNA-mRNA binding, indicating that the silent mutations may potentially contribute to a loss of binding affinity.
Round 2
Reviewer 2 Report
The authors addressed all the comments properly and the manuscript is ready to be considered for publication.